# Learning Goal-Conditioned Policies Offline with Self-Supervised Reward Shaping

**Lina Mezghani**
Meta AI, Inria[*]
linamezghani@fb.com

**Sainbayar Sukhbaatar**
Meta AI

**Piotr Bojanowski**
Meta AI

**Alessandro Lazaric**
Meta AI

**Karteek Alahari**
Inria[*]

**Abstract:**

Developing agents that can execute multiple skills by learning from pre-collected datasets is an important problem in robotics, where online interaction with the environment is extremely time-consuming. Moreover, manually designing reward functions for every single desired skill is prohibitive. Prior works [1, 2] targeted these challenges by learning goal-conditioned policies from offline datasets without manually specified rewards, through hindsight relabeling. These methods suffer from the issue of sparsity of rewards, and fail at long-horizon tasks. In this work, we propose a novel self-supervised learning phase on the pre-collected dataset to understand the structure and the dynamics of the model, and shape a dense reward function for learning policies offline. We evaluate our method on three continuous control tasks, and show that our model significantly outperforms existing approaches [1, 2], especially on tasks that involve long-term planning.

**Keywords:** Offline RL, Self-Supervised Learning, Goal-Conditioned RL

## 1 Introduction

While the goal of realizing general autonomous agents requires mastery of a large and diverse set of skills, achieving this by focusing on each skill individually with standard reinforcement learning (RL) frameworks is prohibitive. This is primarily due to the need for manually designed reward functions and environment interactions for each skill. Unsupervised RL has opened a way for learning agents that can execute diverse abilities without supervision (*i.e.*, hand-crafted rewards), and then be further adapted to downstream tasks through few-shot or zero-shot generalization [3, 4, 5, 6]. However, learning policies with such methods is impractical with real robots as they require millions of interactions when trained online.

Recently, a line of study has emerged that uses pre-collected datasets of trajectories and trains policies offline (*i.e.*, without additional interactions with the environment) [7, 8]. More precisely, given a dataset of reward-free trajectories and a reward function designed to solve a specific task, the agent learns offline by relabeling the transitions in the dataset with the reward function. This setting is particularly relevant in robotics, where data collection is extremely time-consuming: disentangling data collection and policy learning in this context allows for faster policy iteration. However, it would require designing one specific reward function and learning one policy for each individual task.

An important question to scale offline robot learning is therefore to find ways of learning multi-task policies from already collected datasets. Recent works [1, 9, 10], have targeted this problem from a goal-conditioned perspective: given a dataset of previously collected trajectories, the objective is to learn a goal-oriented agent that can reach any state in the dataset. The advantages of this formulation are two-fold: first, it makes it easy to interpret skills, and second it does not require any adaptation at

---

[*]Univ. Grenoble Alpes, Inria, CNRS, Grenoble INP, LJK, 38000 Grenoble, France
Project page: https://linamezghani.github.io/go-fresh

test time. Making this framework unsupervised requires to break free from hand-crafted rewards, as proposed by Chebotar et al. [1], where they learn goal-conditioned policies offline through hindsight relabeling [2]. However, their approach is subject to the pitfall of learning from sparse rewards, and can be inefficient in long-horizon tasks.

In this work, we present a self-supervised reward shaping method that enables building an offline dataset with dense rewards. To this end, we develop a self-supervised learning phase that aims at learning the structure and dynamics of the environment before training the policy. During this phase, we: (i) train a reachability network [11] to estimate the local distance in the state space $\mathcal{S}$, then (ii) extract a set of representative states that covers $\mathcal{S}$, and finally (iii) build a graph on this set to approximate the global distance in $\mathcal{S}$. When training the goal-conditioned policy, we use the graph in two ways: to compute rewards through shortest path distance, and to create transitions of intermediate difficulty on the path to the goal.

We evaluate our method on complex continuous control tasks, and compare it to previous state-of-the-art offline [1, 2] approaches. We show that our graph-based reward method learns good goal-conditioned policies by leveraging transitions from a dataset of past experience with neither any additional interactions with the environment nor manually-designed rewards. Moreover, we show that, contrary to prior work that uses datasets collected with a policy trained with supervised rewards [1], our method allows for learning goal-conditioned policies even from datasets of poor quality, *e.g.* containing trajectories sampled with a random policy. Our work is thus the first to learn goal-conditioned policies from offline datasets without any supervision, as it does not require any hand-crafted reward function at any stage: data collection, policy training and evaluation.

## 2 Related Work

**Goal-conditioned RL.** In its original formulation, goal-conditioned reinforcement learning was tackled by several methods [12, 13, 2, 14]. The policy learning process is supervised in these works: the set of evaluation goals is available at train time as well as a reward function that guides the agent to the goal. Several works propose solutions for generating goals automatically when training goal-conditioned policies, including self-play [15, 16, 17], and adversarial student-teacher policies [18]. A recent line of research [19, 20, 21, 22, 23, 24, 25, 26] focuses on learning goal-conditioned policies in an unsupervised fashion. The objective is to train general agents that can reach any goal state in the environment without any supervision (reward, goal-reaching function) at train time. In particular, Mendonca et al. [25] trains a model-based agent that learns to discover novel goals with an explorer model, and reach them with an achiever policy via imagined rollouts.

**Offline RL.** The data collection technique is an important aspect when studying the training of policies from pre-collected datasets. In this context, the first works assumed access to policies trained with task-specific rewards [27, 28]. More recently, methods proposed to leverage unsupervised exploration to collect datasets for offline RL [7, 8]. In particular, Yarats et al. [7] creates a dataset of pre-collected trajectories, ExoRL, on the DeepMind control suite [29] generated without any hand-crafted rewards. Similar to URLB [30], ExoRL benchmarks a number of exploration algorithms [3, 6, 31, 5], and evaluates the performance of a policy trained on the corresponding offline datasets relabeled with task-specific rewards.

**Multi-task Offline RL.** Recent works proposed to learn multiple tasks from pre-collected datasets, starting with methods [32] that generate goals to improve the offline data collection process in a self-supervised way. This connection has also been studied in the supervised setting [9, 33] and when learning hierarchical policies [10]. In a setting closely related to our work, Actionable Models [1] considers the problem of learning goal-conditioned policies from offline datasets without interacting with the environment, and with no task-specific rewards. They employ goal-conditioned Q-learning with hindsight relabeling [2]. As opposed to their work that relies on learning from sparse rewards, we propose to leverage a self-supervised training stage to densely shape rewards.

## 3 Preliminaries

Let $\mathcal{E} = (\mathcal{S}, \mathcal{A}, P, p_0, \gamma, T)$ define a reward-free Markov decision process (MDP), where $\mathcal{S}$ and $\mathcal{A}$ are state and action spaces respectively, $P : \mathcal{S} \times \mathcal{A} \times \mathcal{S} \to \mathbb{R}_+$ is a state-transition probability

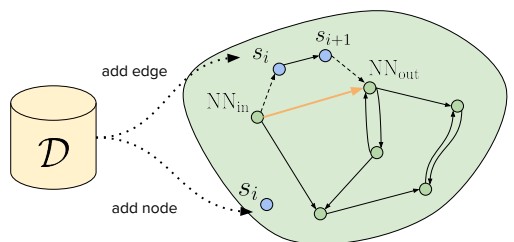

Figure 1: Overview of the graph building algorithm. Given a transition $(s_i, s_{i+1}) \in \mathcal{D}$, we add $s_i$ as node if it is distant enough from existing nodes in the graph. Moreover, we add an edge in the graph between the incoming nearest neighbor of $s_i$ and the outgoing nearest neighbor of $s_{i+1}$.

function, $p_0 : \mathcal{S} \rightarrow \mathbb{R}_+$ is an initial state distribution, $\gamma$ is the discount factor, and $T$ is the task horizon. In the goal-conditioned setting, the objective is to learn a policy $\pi : \mathcal{S} \times \mathcal{G} \rightarrow \mathcal{A}$ that maximizes the expectation of the cumulative return over the goal distribution, where $\mathcal{G}$ denotes the goal space. Here, we make the common assumption that states and goals are defined in the same form, *i.e.*, $\mathcal{G} \subset \mathcal{S}$.

We assume that we have access to a dataset $\mathcal{D}$ of pre-collected episodes generated by using any data collection algorithm in $\mathcal{E}$. Each episode is stored in $\mathcal{D}$ as a series of $(s, a, s')$ tuples, where $s, s' \in \mathcal{S}$ and $a \in \mathcal{A}$. In the general offline formulation introduced by Yarats et al. [7], the dataset $\mathcal{D}$ can be relabeled by evaluating any reward function $r : \mathcal{S} \times \mathcal{A} \rightarrow \mathbb{R}$ at each tuple in $\mathcal{D}$, and adding the resulting tuple $(s, a, r(s, a), s')$ in the relabeled dataset $\mathcal{D}_r$. We can extend this protocol to the goal-oriented setting by considering a goal distribution $p_\mathcal{G}$ in the goal space, and any goal-conditioned reward function $r : \mathcal{S} \times \mathcal{A} \times \mathcal{G} \rightarrow \mathbb{R}$. Given a tuple $(s, a, s')$ in $\mathcal{D}$, we relabel it by sampling a goal $g \sim p_\mathcal{G}$, computing $r(s, a, g)$ and adding the resulting tuple $(s, a, g, r(s, a, g), s')$ in the relabeled dataset $\mathcal{D}_{r, p_\mathcal{G}}$.

Once the relabeled dataset $\mathcal{D}_{r, p_\mathcal{G}}$ is generated, we can learn a goal-conditioned policy by executing any offline RL algorithm. The algorithm runs completely offline, by sampling tuples from $\mathcal{D}_{r, p_\mathcal{G}}$ and without any interaction with the environment. The goal-conditioned policy is then evaluated online in $\mathcal{E}$ on a set of fixed evaluation goals that is not known during training.

## 4 Self-supervised Reward Shaping

We now describe our self-supervised reward shaping method. It comprises three stages that we will detail below. In the first stage, we train a Reachability Network (RNet) [11] on the trajectories in $\mathcal{D}$ to predict whether two states are reachable from one another. The second stage consists in building a directed graph $\mathcal{M}$ whose nodes are a subset of states in $\mathcal{D}$, and edges connect reachable states. We employ the RNet as a criterion to avoid adding similar states to $\mathcal{M}$ so that its nodes cover the states in $\mathcal{D}$ uniformly. The final stage consists in training the goal-conditioned policy on transitions and goals sampled from $\mathcal{D}$. It is trained with dense rewards computed as the sum of a global (based on the graph distance in $\mathcal{M}$) and local (based on the RNet) distance terms. The important aspect of our method is that the whole training only uses trajectories from the pre-collected dataset $\mathcal{D}$ without running a single action in the environment. We now describe each component in more detail.

### 4.1 Reachability network

In order to learn a good local distance between states in $\mathcal{D}$, we adopt an asymmetric version of the Reachability Network (RNet) [11]. The general idea of RNet is to approximate the distance between states in the environment by the average number of steps it takes for a random policy to go from one state to another. We adapted the original formulation with two modifications: first, we use exploration trajectories from $\mathcal{D}$ instead of random trajectories and second, we leverage the temporal direction because a state can be reachable from another without the converse being true. Let $(s_1^a, ..., s_T^a)$ denote a trajectory in $\mathcal{D}$, where $a$ is a trajectory index. We define a *reachability label* $y_{ij}^{ab}$ for each pair of observations $(s_i^a, s_j^b)$ by

$$y_{ij}^{ab} = \begin{cases} 1 & \text{if } a = b \text{ and } 0 \leq j - i \leq \tau_{\text{reach}}, \\ 0 & \text{otherwise}, \end{cases} \qquad \text{for } 1 \leq i, j \leq T, \qquad (1)$$

where the *reachability threshold* $\tau_{\text{reach}}$ is a hyperparameter. The reachability label is equal to 1 *iff* the states are in the same trajectory and the number of steps from $s_i^a$ to $s_j^b$ is below $\tau_{\text{reach}}$, as shown

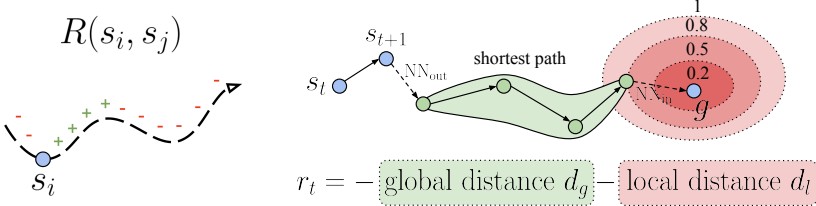

(a) Training labels for RNet          (b) Visualisation of the reward computation

Figure 2: Visualization of our dense reward shaping method. (a) shows how training labels are generated for training the RNet: given a state $s_i$, positive pairs are sampled in the same trajectory within a threshold $\tau_{\text{reach}}$, and the rest of the trajectory forms negative pairs. (b) presents how rewards are implemented as a combination of a global distance term (green), computed with the shortest path in the graph between the outgoing nearest neighbor ($\text{NN}_{\text{out}}$) of the state $s_{t+1}$ and the incoming nearest neighbor of the goal ($\text{NN}_{\text{in}}$), and a local distance term (red) computed using the RNet value between $\text{NN}_{\text{in}}$ and $g$.

in Figure 2a. Note that $y_{ij}^{ab} \neq y_{ji}^{ab}$. We train a siamese neural network $R$, the RNet, to predict the reachability label $y_{ij}^{ab}$ from a pair of observations $(s_i^a, s_j^b)$ in $\mathcal{D}$. The RNet consists of an embedding network $g$, and a fully-connected network $f$ to compare the embeddings, *i.e.*,

$$R(s_i^a, s_j^b) = \sigma \left[ f(g(s_i^a), g(s_j^b)) \right], \tag{2}$$

where $\sigma$ is a sigmoid function. A higher $R$ value indicates two states reachable easily with random walk, so they can be considered close in the environment. More precisely, $R$ takes values in $(0, 1)$ and $s'$ is reachable from $s$ if $R(s, s') \geq 0.5$. RNet is learned in a self-supervised fashion, as the ground-truth labels needed to train the network are generated automatically.

## 4.2 Directed graph

In the next phase, we use trajectories in $\mathcal{D}$ to build a directed graph $\mathcal{M}$ that captures high-level dynamics of the environment, as illustrated in Figure 1. We want the nodes of $\mathcal{M}$ to evenly represent the states in $\mathcal{D}$. This is achieved by filtering the states in $\mathcal{D}$: a state is added to $\mathcal{M}$ only if it is distant enough from all the other nodes in $\mathcal{M}$. More precisely, a state $s \in \mathcal{D}$ is added to $\mathcal{M}$ if and only if

$$R(s, n) < 0.5 \text{ and } R(n, s) < 0.5, \quad \text{for all } n \in \mathcal{M}. \tag{3}$$

Note that we require both the directions to be novel. This filtering avoids redundancy by preventing similar states to be added to the memory. It also has a balancing effect because it limits the number of states that can be added from a certain area even if it is visited by the agent many times in $\mathcal{D}$.

Once the nodes are selected, we connect pairs that are reachable from one to another. To this end, we employ trajectories in $\mathcal{D}$ because they contain actual feasible transitions. Given a transition $s_i \rightarrow s_j$ in $\mathcal{D}$, we add edge $n_i \rightarrow n_j$ if $s_i$ can be reached from node $n_i$ and node $n_j$ can be reached from $s_j$. This way, we have a chain $n_i \rightarrow s_i \rightarrow s_j \rightarrow n_j$ and can assume $n_j$ is reachable from $n_i$. Concretely, we select node $n_i$ to be the incoming nearest neighbor ($\text{NN}_{\text{in}}$) to $s_i$, and $n_j$ to be the outgoing nearest neighbor ($\text{NN}_{\text{out}}$) from $s_j$, *i.e.*,

$$n_i = \text{NN}_{\text{in}}(s_i) = \underset{n \in \mathcal{M}}{\arg\max} \, R(n, s_i), \quad n_j = \text{NN}_{\text{out}}(s_j) = \underset{n \in \mathcal{M}}{\arg\max} \, R(s_j, n). \tag{4}$$

By performing this action over all the transitions in $\mathcal{D}$, we turn $\mathcal{M}$ into a directed graph where edges represent reachability from one node to another.

## 4.3 Distance function for policy training

We then use the obtained directed graph to compute a global distance in the state space. Indeed, RNet predicts reachability between $s_i$ and $s_j$ so we can directly use it as a distance metric

$$d_l(s_i, s_j) = 1 - R(s_i, s_j), \quad \forall s_i, s_j \in \mathcal{S}. \tag{5}$$

However, this reachability metric is confined to a certain threshold, so there is no guarantee that the RNet predictions will have good global properties.

In contrast, the directed graph $\mathcal{M}$ captures high-level global dynamics of the environment. We can easily derive a distance function $d_{\mathcal{M}}(n_i, n_j)$ between any pair of nodes in $\mathcal{M}$ by computing the length of the shortest path in this graph, provided the graph is connected. In practice, we can use a trick to connect the graph if necessary, by adding an edge between the pair of nodes from different connected components with the maximum RNet value. Moreover, we can extend this distance $d_{\mathcal{M}}$ to a global distance function $d_g$ in the state space $\mathcal{S}$ by finding, for any pair $s_i$ and $s_j$ in $\mathcal{S}$ their nearest neighbors in the corresponding direction. More precisely,

$$d_g(s_i, s_j) = d_{\mathcal{M}}(\text{NN}_{\text{out}}(s_i), \text{NN}_{\text{in}}(s_j)), \quad \forall s_i, s_j \in \mathcal{S}. \tag{6}$$

The distance $d_g$ between two states in the state space becomes the length of the shortest path between their respective closest nodes in the graph. This process, summarized in Figure 2b, propagates the good local properties of RNet to get a well-shaped distance function for states that are further away. Since $d_g$ captures global distances while $d_l$ captures local fine-grained distance, we use their combination as a final distance function: $\forall s_i, s_j \in \mathcal{S}, \quad d(s_i, s_j) = d_g(s_i, s_j) + d_l(s_i, s_j)$.

## 4.4 Policy training

The last phase of our method is training the goal-conditioned policy offline. Here, we create an offline replay buffer $\mathcal{B}$ that is filled with relabeled data. We randomly sample a transition $(s_t, a_t, s_{t+1})$ from $\mathcal{D}$ as well as a goal $g$ and relabel the transition with reward $r_t = -d(s_{t+1}, g)$. We then push the relabeled transition $(s_t, a_t, g, r_t, s_{t+1})$ to $\mathcal{B}$. In order to create a curriculum that artificially guides the agent towards the goal, we experimented with two different transition augmentation techniques:

**Sub-goal augmentation.** Let $(s_t, a_t, g, r_t, s_{t+1})$ denote a relabeled transition and $(n_0, ..., n_{P-1})$ the shortest path in the graph $\mathcal{M}$ between $n_0 = \text{NN}_{\text{out}}(s_t)$ and $n_{P-1} = \text{NN}_{\text{in}}(g)$. The augmentation technique consists in adding to the replay buffer every transition $(s_t, a_t, n_i, r_t^i, s_{t+1})$ for all $i \in \{0, P-1\}$, where $r_t^i = -d(s_{t+1}, n_i)$. In other words, given a transition $(s_t, a_t, s_{t+1})$ and a goal $g$ from $\mathcal{D}$, we push to the replay buffer a set of relabeled transitions with all goals on the shortest path from $s_t$ to $g$ (and their corresponding rewards).

**Edge augmentation.** Similar to the subgoal augmentation technique, we consider a relabeled transition $(s_t, a_t, g, r_t, s_{t+1})$ and the associated shortest path $(n_0, ..., n_{P-1})$. This time, we keep the same goal $g$ for every augmented transition, but for every edge $(n_{i-1}, n_i), i \in \{1, P-1\}$, we add the relabeled transition $(s_t^i, a_t^i, g, r_t^i, s_{t+1}^i)$ to $\mathcal{B}$ where $(s_t^i, a_t^i, s_{t+1}^i) \in \mathcal{D}$, $\text{NN}_{\text{out}}(s_t^i) = n_{i-1}$, $\text{NN}_{\text{in}}(s_{t+1}^i) = n_i$ and $r_t^i = -d(s_t^i, g)$. Note that the existence of such a transition in $\mathcal{D}$ is guaranteed by construction: an edge is added to the graph from one node to another *iff* there exist a transition in $\mathcal{D}$ whose corresponding nearest neighbors are these two nodes (in the same order).

Once the replay buffer $\mathcal{B}$ is filled, the goal-conditioned policy can be trained using any off-policy algorithm. In our implementation, we chose Soft Actor-Critic [34], as it is known to require few hyper-parameter tuning, and is widely used in the literature.

## 5 Experiments

### 5.1 Environments & data collection

We perform experiments on three continuous control tasks with state-based inputs.

**UMaze [35].** The first environment, shown in Figure 3a, is a two-dimensional U-shaped maze with continuous action space and a fixed initial position. We generate the training data for this environment by deploying a random policy with randomized start position in the maze. We collect 10k trajectories of length 1k. We evaluate the goal-conditioned agent by giving the agent a goal sampled at random in the environment and computing the final euclidean distance to the goal.

**RoboYoga Walker [25].** Introduced by Mendonca et al. [25], the challenging RoboYoga benchmark is based on the Walker domain of the DeepMind control suite [29], and consists of 12 goals

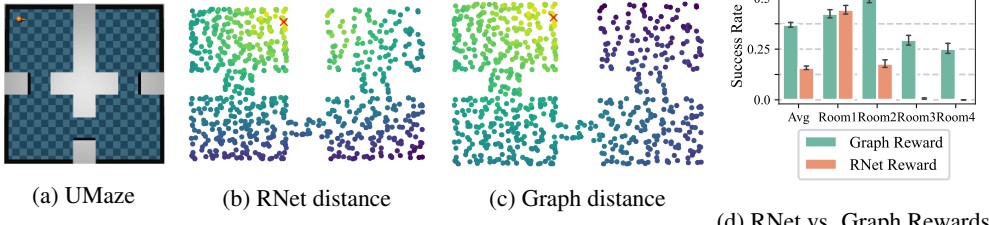

(a) UMaze     (b) RNet distance     (c) Graph distance

(d) RNet vs. Graph Rewards

Figure 3: (a) UMaze environment, Heatmap of rewards computed with RNet (b) and graph (c) distances, and (d) Performance of the goal-conditioned policy trained with RNet and graph-based rewards on UMaze. In (b) and (c), high rewards are shown in yellow, and low rewards in black.

that correspond to body poses inspired from yoga (*e.g.* lying down, raising one leg or balancing). We consider the state-based version of the task, and use the task-agnostic dataset from Yarats et al. [7] generated with an unsupervised exploration policy. It contains 10k trajectories of length 1k obtained by deploying the "proto" [5] algorithm in the Walker domain. The success metric of the evaluation policy is assessed by the pose of the humanoid at the end of the episode.

**Pusher [20].** We also apply our method on *Pusher*, a realistic robotic environment shown in Figure 7 (left), where a robot arm (red) needs to push a puck (blue) to a specified location on a table. To build the offline dataset, we generated 10k random trajectories of length 200. Similar to prior works [20, 22, 26], we generated 500 goals at random in the state space, and we measured the performance as the final Euclidean distance between the puck and its target location.

## 5.2 Ablation & design choices

We first show that the graph structure is necessary for long-term planning. Then, we explain the importance of the directness of the graph on tasks with asymmetric behaviours. Finally, we show the impact of transition augmentation techniques when labeling data for the goal-conditioned policy.

**Necessity of graph-based rewards.** An important component of our method is the construction of the graph $\mathcal{M}$ that enables computing a distance with good global properties. To empirically validate this hypothesis, we performed a comparison between the goal-conditioned policy trained with RNet rewards (*i.e.*, by using the distance $d_l$ from equation (5)) and the one trained with both distance terms as reward. We run this experiment on the UMaze environment, and show results in Figure 3d. We note that the model trained with graph rewards outperforms the one trained with RNet rewards overall, particularly for distant goals (ie. rooms 3 and 4). We also notice that the model trained with RNet rewards is slightly better for goals that are close to the initial position. This highlights the fact that RNet is good at estimating local distances. The qualitative visualization in Figure 3b & 3c confirms this observation, as it shows low values between states in the first and fourth rooms.

**Importance of graph directness.** We then investigate the importance of the asymmetry of the RNet and the directness of the graph. To this end, we implement an undirected version of our method where the RNet is symmetric and the graph is undirected. All other components of our method are unchanged. First, we compare the performance of both variants in the UMaze task in Figure 4a, and note that asymmetric RNet and directed graph in our approach significantly improve the goal-conditioned policy performance ($+11\%$ on success rate), especially on goals close to the initial location, *i.e.*, goals in rooms 1 and 2. We then analyze qualitative visualizations of the shortest path in the undirected and directed graphs in the RoboYoga task, as shown in Figure 4b. In the undirected case, the humanoid defies the laws of gravity and is encouraged to stand its head by flipping backwards, which might be extremely difficult, or even infeasible. In the directed case, the shortest path fosters the agent to first get back on its legs, and then lean forward. In this exemple, the gravity makes the dynamics of the environment non-symmetric and non-fully reversible, which justifies the directed formulation described in our method.

**Transition sampling strategy.** As a final ablation study, we study the utility of the transition augmentation techniques described in subsection 4.4. We evaluate four possible variants of our method:

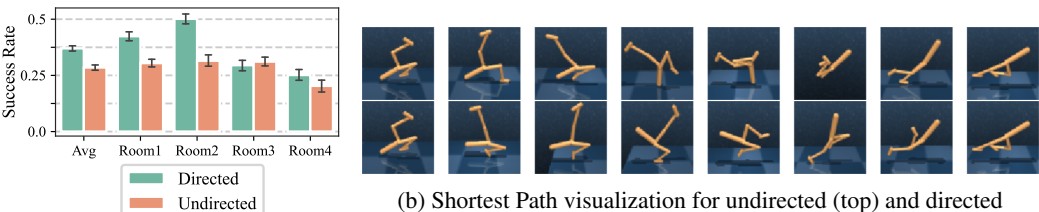

(a) Comparison on the UMaze task

(b) Shortest Path visualization for undirected (top) and directed (bottom) graphs

Figure 4: Importance of graph directness on (a) the UMaze task and (b) the RoboYoga Walker task.

(i) without any augmentation, (ii) with edge augmentation only, (iii) with subgoal augmentation only, and (iv) with both augmentations. We execute this experiment on the RoboYoga task, and show results in Figure 6b. We observe that both of the augmentation techniques improve the performance of the goal-conditioned agent, with subgoal augmentation showing greater improvement. Moreover, we note that combining both augmentations improves the performance further. For the reminder of the experiments, we use both these augmentation techniques.

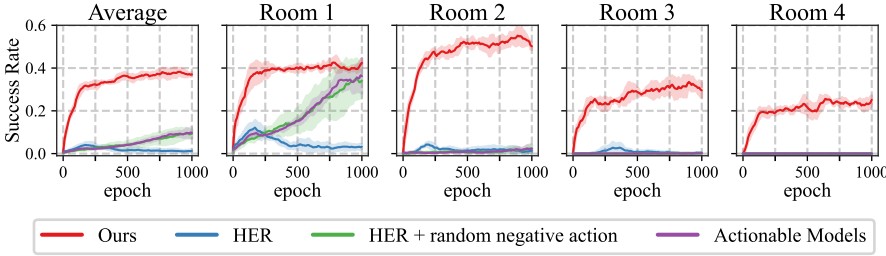

Figure 5: Performance on the UMaze task. We show the success rate for goals sampled at random in each of the four rooms, as well as the average over all rooms.

## 5.3    Comparison to prior work

**Baselines.** We compare our method to prior work on unsupervised goal-conditioned policy learning. We perform an apples-to-apples comparison by implementing the baselines using the same learning framework as our method, and changing the reward relabeling process. We compare with the following baselines:

- **Hindsight Experience Replay [HER] [2]** This is a re-implementation of the standard unsupervised RL technique, adapted to the offline setting. More precisely, we relabel sub-trajectories from $\mathcal{D}$ with a sparse reward, which is equal to 1 only for the final transition of the sub-trajectory, and 0 everywhere else. Following Chebotar et al. [1], we also label sub-trajectories with goals sampled at random in $\mathcal{D}$ and zero reward.
- **HER [2] with random negative action** is a variant of HER where, for a transition in $\mathcal{D}$ we sample an action uniformly at random in the action space and label it with zero reward. This helps overcoming the problem of over-estimation of the Q-values for unseen actions mentioned in Chebotar et al. [1].
- **Actionable Models [1]** This approach is based on goal-conditioned Q-learning with hindsight relabeling. We re-implemented the goal relabeling procedure that uses the Q-value at the final state of sub-trajectories in $\mathcal{D}$ to enable goal chaining, as well as the negative action sampling trick.

**Comparison on UMaze.** We compare our method to the baselines on the UMaze task, and show results in Figure 5. We observe that our model outperforms all baselines overall, and shows greater improvements on challenging goals that are far from the initial position. Interestingly, we note that Actionable Models reaches goals in the first room only. This confirms the intuition that sparse rewards make it difficult for the policy to learn long-horizon tasks.

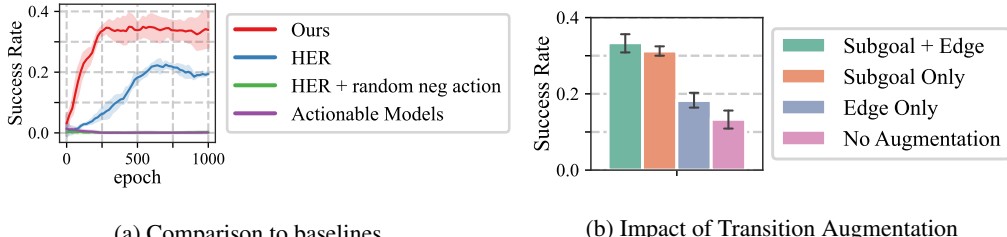

(a) Comparison to baselines

(b) Impact of Transition Augmentation

Figure 6: Performance on the RoboYoga Walker task

**Comparison on RoboYoga Walker.** In a second experiment, we compare our method to baselines on the RoboYoga task, as shown in Figure 6a. Here again, our method outperforms prior work, and Actionable Models does not make any significant improvement over HER. The results broken down by goal are shown in the supplementary material. Overall these results suggest that our dense reward shaping method allows for faster and more robust offline goal-conditioned policy training.

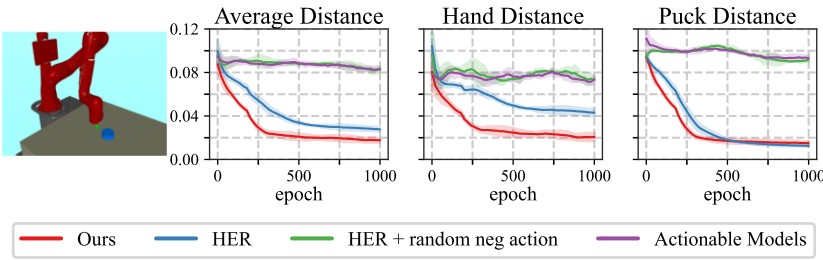

Figure 7: Performance on the Pusher task (lower is better). We report the final average, hand, and puck distance to the goal for our model and all baselines.

**Comparison on Pusher.** As a final experiment, we compared our method to prior work on a realistic robotic environment, as shown in Figure 7. Our policy trained offline is evaluated by sampling a goal at random in the state space, and measuring three different metrics: (i) the *hand distance*, which corresponds to the final distance between the end of the robot arm and the target, (ii) the *puck distance*, which measures the distance between the final puck location and the target, and (iii) the *average distance*, the average of the first two metrics. Our method outperforms the baselines on this task, and our goal-conditioned agent is able to sequentially place the puck at the goal location, and then place the hand at its target location. On the contrary, **HER [2]** places the puck at the target location with a performance similar to our method, but lacks precision on the hand location.

## 6   Conclusion: Summary and Limitations

We proposed a method for learning multi-task policies from pre-generated datasets in an offline and unsupervised fashion, *i.e.*, without requiring any additional interaction with the environment, nor manually designed rewards. Our method leverages a self-supervised stage that aims at learning the dynamics of the environment from the offline dataset, and that allows for shaping a dense reward function. It shows significant improvement over prior works based on hindsight relabeling, especially on long-horizon tasks, where dense rewards are crucial for learning a good policy.

The main limitation of our method is that it relies on the availability of a pre-collected dataset of trajectories, with a sufficiently large coverage of the state space for proper policy learning. Although such data can be already available, as for the RoboYoga Walker task, or that offline dataset collection could be done with random policies, as we did on the UMaze and Pusher tasks, this step can be challenging for other environments. Another limitation is that we evaluated our method exclusively on simulated environments, and we did not perform any experiments on real robots, for which pre-collected dataset with expert demonstrations can be available [1].

**Acknowledgments**

Karteek Alahari is supported in part by the ANR grant AVENUE (ANR-18-CE23-0011).

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
