# OpenReview forum: "Learning Goal-Conditioned Policies Offline with Self-Supervised Reward Shaping"
_robot-learning.org/CoRL/2022/Conference — CoRL 2022 Poster_

### Official Review · Reviewer_4D4F · 2022-07-31

**Originality:** Good
**Technical Quality:** Good
**Clarity Of Presentation:** Very Good
**Impact:** 3

**Recommendation:**

Weak Accept: I recommend accepting the paper, but will not argue for my recommendation if the majority of other reviewers have a different opinion.

**Summary:**

This paper proposes a method for learning goal-conditioned policies from reward-free offline datasets. To this end, the method first learns a reachability network, constructs a graph over a state space, and re-label and augment the dataset using the reward defined as a sum of distance measured using the graph and the reachability network. Then the off-policy goal-conditioned RL agent is trained using the dataset. The method is evaluated on UMaze and RoboYoga environments.

**Issues:**

- Additional robotic experiments, especially manipulation experiments that would enable direct comparison with the baseline
- Improved presentation
- Experiments with high-dimensional observations

**Quality Of The Limitations Section:**

Limitations section not present

**Reviewer Expertise:**

4: The reviewer is confident but not absolutely certain that the evaluation is correct

**Robotics Focus:**

Highly relevant to robotics but no hardware experiments

**Strengths And Weaknesses:**

Strengths
- Method is a bit complicated but makes sense and shown to be effective on various tasks
- Experiments are well conducted to show the benefit of the proposed techniques with ablations

Weaknesses
- Experiments are conducted on relatively simple tasks -- Performance on more complex tasks, especially manipulation tasks, could further support the claim of the proposed method for learning distance in state space. As a baseline (i.e., Actionable Models) shows its effectiveness on various robotic experiments, I would expect this could also be the case for the proposed method, but explcitly showing this would be required for making the paper strong and more relevant to robotics.
- In terms of presentation, Figure 1 is helpful for understanding high-level concept but more figures for detailed methods would help understanding in detail, especially for augmentations, because notations are a bit complex. For notation, introducing superscripts as a notation for trajectory index and also introducing subscript differently for augmentations makes it a bit difficult to understand what's going on.
- I would not say that this is necessary; but it would be interesting to see how the proposed method performs with high-dimensional observations (pixels) by training a reachability network on pixel observations. At least including the discussion on this extension in the paper could be interesting and helpful for future researches.

Minor
- It's awkward to say to call a robot used in DMC Walker as 'humanoid'
- Figure 3: information about what does each color mean is missing

**Summary Of Recommendation:**

I would recommend this paper to be accepted because the method is intuitive and results are strong. But it's difficult to argue strong acceptance because there is a room for improvement, e.g., including additional robotic experiments and improving presentation.

---

> ### Author Response · Authors · 2022-08-25
> **Response to Reviewer 4D4F**
>
> We thank the reviewer for their comments and address the points raised below. A rebuttal document, with additional material is attached to this response.
>
> We agree with the reviewer that a stronger validation in more robotic-plausible environments is necessary. To address this issue, we performed additional experiments on a robotics task (Pusher), in which a realistic robot arm needs to push a puck to a specified location on a table. This task was widely studied in prior works [16, 19, 20, 3] and is a popular setup for more realistic robotic experiments. The results for these experiments are available in the rebuttal document in Figure 1 and they confirm the findings in the other experiments presented in the paper, where our method consistently performs better than baselines, including HER, which is known to be a strong algorithm for this type of problem.
>
> We took into consideration the suggestion of the reviewer to add more figures to improve the clarity of the presentation, and added Figure 2 in Section 3 of the rebuttal document. Moreover, we plan to add another figure to illustrate augmentation techniques in Section 4.4 in the revised version of the paper.
>
> We also agree with the reviewer that it would be interesting to discuss experiments on tasks with pixel observations. However, the purpose of the paper is to show that we can train meaningful goal-based policies by learning a representation that preserves the dynamics of the environment. Working on image inputs would simply add a perceptual representation learning problem which would be a confounding factor with respect to the main focus of learning proper distances between states.
>
> Finally, we thank the reviewer for the comment on the limitations, and have added it in the rebuttal document (Section 2).
>
> [3] Rethinking goal-conditioned supervised learning and its connection to offline rl. Yang et al. 2022 \
> [16] Skew-fit: State-covering self-supervised reinforcement learning. Pong et al. ICML 2020 \
> [19] Discovering and Achieving Goals via World Models. Mendonca et al. NeurIPS 2021 \
> [20] Walk the random walk: Learning to discover and reach goals without supervision. Mezghani et al. ICLRW 2022

---

> > ### Comment · Reviewer_4D4F · 2022-08-26
> > **Thanks for the response**
> >
> > Thanks for the response! It's still a bit difficult for me to see that Pusher task is approaching the difficulty of robot learning tasks considered in a prior work (Actionable Models). And it's not clear why prior work that works on more complex tasks largely underperforms even HER. Hence I'm keeping the score of 'weak accept'. It would be very nice if additional results will be available in the camera-ready version, if accepted.

---

> > > ### Author Response · Authors · 2022-08-26
> > > **Thank you for the suggestion**
> > >
> > > Thank you for the suggestion. We will look into additional experiments for the final draft.

---

### Official Review · Reviewer_nCHc · 2022-07-31

**Originality:** Good
**Technical Quality:** Good
**Clarity Of Presentation:** Very Good
**Impact:** 3

**Recommendation:**

Weak Accept: I recommend accepting the paper, but will not argue for my recommendation if the majority of other reviewers have a different opinion.

**Summary:**

This paper is concerned with unsupervised goal-conditioned policy learning. Manually designing reward functions for every single desired skill is very expensive. The authors propose a novel self-supervised learning phase on the pre-collected dataset to understand the structure and the dynamics of the model, and shape a dense reward function for learning policies offline. The proposed algorithm is benchmarked on a gridworld environment UMaze and RoboYoga walker environment from DMControl.

**Issues:**

My main issue is with the toy experiment settings. Please see "Weakness" section.

In addition, the limitation section is not really there in Sec. 6, requires a lot more expansion.

**Quality Of The Limitations Section:**

Limitations are not well addressed

**Reviewer Expertise:**

4: The reviewer is confident but not absolutely certain that the evaluation is correct

**Robotics Focus:**

Relevant but unlikely to deploy to hardware in near future

**Strengths And Weaknesses:**

**Strengths**:

* This paper is well-written and easy to follow. The motivation of goal-conditioned offline policy learning is well explained.
* To my knowledge, the proposed method of combining Reachability Network (RNet) and a graph-based distance (reward) metric is novel. The method and details are also well-explained.

**Weaknesses**:

The proposed algorithm is benchmarked on 2 environments, UMaze (a toy gridworld) and RoboYoga (DMControl, another artificially constructed environment). While the experimental results are informative, neither environments have practical value or concrete applications beyond a simple demonstration. For example, the Walker bot in RoboYoga does not have a real counterpart and cannot be realized on available hardware. In prior works such as LEXA [1], empirical results are shown on simulated robotics task like RoboBins and RoboKitchen that involve realistic robot grippers. Since this is a robot learning conference, it is appropriate to show more meaningful and useful robotic tasks.

Furthermore, limitations are not well discussed in Sec. 6. The current statement is insufficient.

[1] Discovering and Achieving Goals via World Models. Mendonca et al. 2021.


**Summary Of Recommendation:**

I am willing to consider raising score if the proposed method can achieve convincing results on simulated robotic tasks that are more meaningful and practically useful.

---

> ### Author Response · Authors · 2022-08-25
> **Response to Reviewer nCHc**
>
> We thank the reviewer for their comments and address the points raised in the following. A rebuttal document, with additional material is attached to this response.
>
> While the tasks we investigate in the paper serve the purpose of studying the effectiveness of the method, we understand that a stronger validation in more robotic-plausible environments is necessary. To address this issue, we performed additional experiments on a robotics task (Pusher), in which a realistic robot arm needs to push a puck to a specified location on a table. This task was widely studied in prior works [16, 19, 20, 3] and is a popular setup for more realistic robotic experiments. The results for these experiments are available in the rebuttal document in Figure 1 and they confirm the findings in the other experiments presented in the paper, where our method consistently performs better than baselines, including HER, which is known to be a strong algorithm for this type of problem.
>
> We also acknowledge that the paper needs a better discussion on the limitations, and have added it in the rebuttal document (Section 2).
>
> [3] Rethinking goal-conditioned supervised learning and its connection to offline rl. Yang et al. 2022 \
> [16] Skew-fit: State-covering self-supervised reinforcement learning. Pong et al. ICML 2020 \
> [19] Discovering and Achieving Goals via World Models. Mendonca et al. NeurIPS 2021 \
> [20] Walk the random walk: Learning to discover and reach goals without supervision. Mezghani et al. ICLRW 2022

---

> > ### Comment · Reviewer_nCHc · 2022-08-25
> > **Raising score**
> >
> > Thanks for the additional experiment and clarifications. I will raise my score to weak accept when the portal unlocks (cannot edit my evaluation for now).

---

### Official Review · Reviewer_iP9J · 2022-08-01

**Originality:** Good
**Technical Quality:** Good
**Clarity Of Presentation:** Very Good
**Impact:** 3

**Recommendation:**

Weak Accept: I recommend accepting the paper, but will not argue for my recommendation if the majority of other reviewers have a different opinion.

**Summary:**

The paper addresses the problem of learning long-horizon tasks from offline (pre-collected) datasets of behaviors. The paper proposes to learn the structure of the task by learning a global distance in the state space of the MDP by building a graph prior to the offline training. Similar to prior works, using hindsight experience replay, the method constructs a dense reward signal to be used for goal reaching tasks with sparse rewards.

**Issues:**

Line 164-165: "we can use a trick to connect the graph if necessary by adding ...": This is very unclear to understand what the problem is and what the solution is.

Line 186 (Edge augmentation): It is  unclear where the action a_t^i comes from. A new transition (augmented one) is found by finding s being an incoming state to the node n_{i-1}, and s' being an outgoing state for the node n_i. Therefore, to connect s to s' we need more than one action, most likely a sequence of actions.

Line 229 (Importance of graph directness): It is obvious that we need asymmetric graph for the given task. A symmetric graph would work for tasks like 2D maze in which actions are reversible. Considering this, I don't understand the purpose of the ablation other than answering a very obvious question.

**Quality Of The Limitations Section:**

Limitations are not well addressed

**Reviewer Expertise:**

4: The reviewer is confident but not absolutely certain that the evaluation is correct

**Robotics Focus:**

Highly relevant to robotics but no hardware experiments

**Strengths And Weaknesses:**

Strength:
* The paper proposes a novel method that can solve long-horizon RL tasks provided offline datasets of state-action transitions.

Weaknesses:
* The paper does not compare to most recent self-supervised offline RL algorithms such as "RETHINKING GOAL-CONDITIONED SUPERVISED LEARNING AND ITS CONNECTION TO OFFLINE RL" (reference [3]).
* The experimental results are limited to only two tasks.
* The method does not seem to scale to vision-based policy training tasks since the graph cannot be constructed over image space.
* Please also see the issues.

**Summary Of Recommendation:**

The paper introduces a novel method to solve challenging long-horizon tasks in offline RL. The experimental results and the ablations show benefits of the method on two long-horizon offline RL tasks. However, one important baseline is missing and, since the experiments are done in simulation, one can expect results on more than two tasks.

---

> ### Author Response · Authors · 2022-08-25
> **Response to Reviewer iP9J**
>
> We thank the reviewer for their comments and address the points raised below. A rebuttal document, with additional material is attached to this response.
>
> > “The paper does not compare to most recent self-supervised offline RL algorithms such as [3]”
>
> we would like to highlight our main assumption that external rewards are not available, contrary to the afore-mentionned work [3]  which assumes access to a sparse reward upon reaching goal. To the best of our knowledge, the only prior work that makes the same assumption is Actionable Models [1], which we compare against  in all of our experiments.
>
> > “The experimental results are limited to only two tasks.”
>
> To address this issue, we performed additional experiments on a robotics task (Pusher), in which a realistic robot arm needs to push a puck to a specified location on a table. This task was widely studied in prior works [16, 19, 20, 3] and is a popular setup for more realistic robotic experiments. The results for these experiments are available in the rebuttal document in Figure 1 and they confirm the findings in the other experiments presented in the paper, where our method consistently performs better than baselines, including HER, which is known to be a strong algorithm for this type of problem.
>
> > “The method does not seem to scale to vision-based policy training tasks since the graph cannot be constructed over image space.”
>
> In theory, there is no clear reason that would make our method fail on image inputs, as the Reachability Network should learn low-level representations that make the graph construction feasible. This has been done in the paper that introduces the Reachability Network [5], and in follow-up works [20]. In practice though, we agree that there may be technical difficulties to scale our method to pixel inputs, and it would be interesting to study the feasibility of it. However, the purpose of this paper is to show that we can train meaningful goal-based policies by learning a representation that preserves the dynamics of the environment. Working on image inputs would simply add a perceptual representation learning problem which would be a confounding factor with respect to the main focus of learning proper distances between states.
>
> > “Line 164-165: we can use a trick to connect the graph if necessary by adding ...,This is very unclear to understand what the problem is and what the solution is”
>
> We acknowledge that this formulation can be improved for clarity. Here, we mean that if the graph is not connected, there exists at least a pair of nodes for which the length of the shortest path in the graph is not defined. This can result in computation issues when labeling the replay buffer transitions with rewards. The trick to connect the graph is to select one of the connected components in the graph, and to compute the RNet values between every node in this connected component and the rest of the nodes in the graph, and create an edge between pairs of nodes with maximum RNet value. We repeat this process until the graph is connected. We will clarify this procedure in the revised version of the paper.
>
> > “Line 186 (Edge augmentation): It is unclear where the action a_t^i comes from.”
>
> The augmented transition is selected in the offline dataset among existing transitions. We are therefore sure that there exists an action that goes from s to s', which is the action from the corresponding transition.
>
> > “Line 229 (Importance of graph directness): It is obvious that we need an asymmetric graph for the given task”
>
> Please note that there are several methods that symmetrize the dynamics. For example, works based on Laplacian representations [A, B], where the features are the Eigenvectors of the Laplacian of the graph (obtained by a random walk), symmetrize the dynamics and do not take into account the directedness of the environment. The purpose of this experiment was therefore to show empirical evidence that taking into account the directedness of the dynamics was crucial for some tasks.
> We also acknowledge that the discussion on limitations is insufficient, and have added a section in the rebuttal document (Section 2).
>
> [1] Actionable models: Unsupervised offline reinforcement learning of robotic skills, Chebotar et al., ICML 2021 \
> [3] Rethinking goal-conditioned supervised learning and its connection to offline rl. Yang et al. 2022 \
> [5] Episodic Curiosity through Reachability, Savinov et al., ICLR 2018 \
> [16] Skew-fit: State-covering self-supervised reinforcement learning. Pong et al. ICML 2020 \
> [19] Discovering and Achieving Goals via World Models. Mendonca et al. NeurIPS 2021 \
> [20] Walk the Random Walk: Learning to Discover and Reach Goals Without Supervision, Mezghani et al., ICLRW 2022 \
> [A] Proto-value functions: Developmental reinforcement learning, Mahadevan, ICML 2005 \
> [B] Towards better laplacian representation in reinforcement learning with generalized graph drawing, Wang et al., ICML 2021

---

### Official Review · Reviewer_4myo · 2022-08-05

**Originality:** Very Good
**Technical Quality:** Very Good
**Clarity Of Presentation:** Very Good
**Impact:** 4

**Recommendation:**

Weak Accept: I recommend accepting the paper, but will not argue for my recommendation if the majority of other reviewers have a different opinion.

**Summary:**

The paper proposes a novel method to learn from offline datasets. The paper proposes to first learn an RNet, and then distill the RNet predictions onto a DAG to obtain self supervised rewards from the datasets in a self-supervised manner, which does not require reward specification on any sort, at any stage of the pipeline. The paper evaluated the methods on two continous control tasks and improves upon prior self-supervised reward learning methods like HER.

**Issues:**

Please see weaknesses above

**Quality Of The Limitations Section:**

Limitations are not well addressed

**Reviewer Expertise:**

3: The reviewer is fairly confident that the evaluation is correct

**Robotics Focus:**

Relevant but unlikely to deploy to hardware in near future

**Strengths And Weaknesses:**

Strengths:
- The employment of both Rnet and the graph distance seems like a novel and exciting approach.
- The subgoal and edge augmentation are very interesting and practical solutions during policy training, since the
- Results on YogaWalker are good.
- The ablation is thorough and discusses all different variants of their proposed algorithm.
- Fig 1 is extremely well done.

Weaknesses:
- The way the paper is presented, it seems to be mostly for discrete state spaces to construct the graph. But the results shown on YogaWalker is obviously a cts state space. I am confused how the paper constructed the graph for a cts space.
- I would ideally like to see more experiments with cts state space tasks.
- I would like to see the paper explore real robotics tasks, if possible.
- RNet predicted distance in fig 3 a seem very wrong. Rnet is seemingly overfitting to the xy position distance not the actual map of how "reachable" the far ends of the maps are. I am not convinced the RNet is learning anything meaningfully correct, and the graph is doing the heavy lifting. To me it seems like R is not learning anything particularly meaningful, and the distillation to the DAG is somewhat lucky due to the hill-climbing way of learning.
- Using DAGs are a sensible choice, but the paper doesn't empirically show it in a convincing manner.
- "we leverage the temporal direction because a state can be reachable from another without the converse being true", i would be interested in a short analysis of if the fwd-bwd models are consistent, and some intuition about if it makes sense for them to be consistent. For most robotic tasks, I can imagine the A->B and B->A is equally reachable, but I am not sure if that holds for some of the very specific tasks considered.
- Intuitively, the method would perform well on similar tasks as the original HER paper. I would like to see some results on manipulation tasks.
- NIT: typically ablation should follow the main experiments / comparisons with the SOTA



**Summary Of Recommendation:**

The paper presents a novel, and very interesting approach to learn from offline datasets, using a self-supervised approach, which employs learning a graph to measure the reachability between two states. Overall, the method is extremely interesting, but the experimental evaluation is not sufficiently thorough.

---

> ### Author Response · Authors · 2022-08-25
> **Response to Reviewer 4myo**
>
> We thank the reviewer for their comments, and clarify the points raised in the following. A rebuttal document, with additional material is attached to this response.
>
>
> > “it seems to be mostly for discrete state spaces to construct the graph”
>
> we would like to clarify the fact that our proposed method constructs a graph whose set of nodes is a subset of the state spaces, which can be either discrete or continuous. Our experiments were all performed in environments with continuous state spaces.
>
>
> > “I would ideally like to see more experiments with cts state space tasks, real robotics tasks [and] on manipulation tasks”
>
> We agree with the reviewer that a stronger validation in more robotic-plausible environments is necessary. To address this issue, we performed additional experiments on a robotics task (Pusher), in which a realistic robot arm needs to push a puck to a specified location on a table. This task was widely studied in prior works [16, 19, 20, 3] and is a popular setup for more realistic robotic experiments. The results for these experiments are available in the rebuttal document in Figure 1 and they confirm the findings in the other experiments presented in the paper, where our method consistently performs better than baselines, including HER, which is known to be a strong algorithm for this type of problem.
>
>
> > “To me it seems like R is not learning anything particularly meaningful, and the distillation to the DAG is somewhat lucky due to the hill-climbing way of learning.”
>
> We disagree on this point. As shown in the heatmap of Figure 3.a) the RNet is learning a distance which is meaningful for nearby states, even though it is not consistent for far-away states. The distillation to the graph is therefore due to this good local distance being propagated globally with the graph building process.
>
>
> > “Using DAGs is a sensible choice, but the paper doesn't empirically show it in a convincing manner.”
>
> We empirically validated the use of the directed graph in two different experiments. First in Figure 2.b), we compare the performance of the goal-conditioned policy trained using both graph and RNet rewards, with the one trained with RNet rewards only on the UMaze task, and we show that the graph rewards are necessary for reaching states that are far from the start location. Second, in Figure 4 of the Supplementary material we show that the graph directedness is crucial on this same task.
>
>
> > “For most robotic tasks, I can imagine the A->B and B->A are equally reachable, but I am not sure if that holds for some of the very specific tasks considered”
>
> We would like to clarify the fact that, even in robotics it is true that, if we can go from A to B, oftentimes we can go from B to A, yet it may not happen in one single step. For instance, consider a point mass falling down a slope. While we may observe the mass moving from the top to the bottom in one single step and it may indeed be possible to climb back to the top, this may take many steps depending on the available actions. In this case, the dynamics is "globally" reversible, but connecting the bottom of the slope to the top with one single edge would lead to poor reward shaping and learning performance. This type of dynamics is indeed often present in the RoboYoga Walker environment we used in our experiments.
>
> We also thank the reviewer for the comment on the limitations section. We added it in the rebuttal document (Section 2).
>
> [3] Rethinking goal-conditioned supervised learning and its connection to offline rl. Yang et al. 2022 \
> [16] Skew-fit: State-covering self-supervised reinforcement learning. Pong et al. ICML 2020 \
> [19] Discovering and Achieving Goals via World Models. Mendonca et al. NeurIPS 2021 \
> [20] Walk the random walk: Learning to discover and reach goals without supervision. Mezghani et al. ICLRW 2022

---

### Meta-Review · Area_Chair_Lm6Z · 2022-08-13

**Recommendation:** Accept (Poster)
**Confidence:** 5

**Metareview:**

# Strengths
- The approach is interesting and the idea of combining reachability networks with graph-based distances is novel and promising

# Weaknesses
- The evaluation is fairly limited wrt the chosen scenarios.
- There are some concerns regarding practicability and scalability to real-world robotic tasks.
- The paper would strongly benefit from additional evaluations on manipulations tasks comparable to the tasks conducted by prior work.
## Limitations
The paper does not discuss the limitations of the proposed approach. Such as discussion is however required by the conference.

# Post-Rebuttal Update
The authors did address the concerns of the reviewers and even evaluated the approach on an additional experimental task.
While I appreciate the work done in the rebuttal phase I expect that the additional experiments in the rebuttal document as well as the clarifications provided in the discussion with the reviewers will be added to the camera-ready version of the paper.
Furthermore, the limitations in the rebuttal document are still very superficial and don't provide any valuable insight into the constraints and downsides of the method. I strongly encourage the authors to provide an insightful limitation section for the camera-ready version.

**Best Paper Nomination:**

No

---

> ### Author Response · Authors · 2022-08-25
> **Response to Meta Review**
>
> We thank the area chair for their meta-review. We apologize for the late response, we wanted to run additional experiments before replying.
>
> We agree with the area chair  that a stronger validation in more robotic-plausible environments is necessary. To address this issue, we performed additional experiments on a robotics task (Pusher), in which a realistic robot arm needs to push a puck to a specified location on a table. This task was widely studied in prior works [16, 19, 20, 3] and is a popular setup for more realistic robotic experiments. The results for these experiments are available in the rebuttal document (attached to this response) in Figure 1 and they confirm the findings in the other experiments presented in the paper, where our method consistently performs better than baselines, including HER, which is known to be a strong algorithm for this type of problem.
>
> Finally, we thank the area chair for the comment on the limitations, and have added it in the rebuttal document (Section 2).
>
> [3] Rethinking goal-conditioned supervised learning and its connection to offline rl. Yang et al. 2022 \
> [16] Skew-fit: State-covering self-supervised reinforcement learning. Pong et al. ICML 2020 \
> [19] Discovering and Achieving Goals via World Models. Mendonca et al. NeurIPS 2021 \
> [20] Walk the random walk: Learning to discover and reach goals without supervision. Mezghani et al. ICLRW 2022